# Solid Target System with In-Situ Target Dissolution

**William Z. Gelbart [1],\* and Richard R. Johnson [2]**

1    Advanced Systems Designs Inc., Garden Bay, BC V0N 1S1, Canada
2    Deptment of Physics and Astronomy, University of British Columbia, Vancouver, BC V6T 1Z4, Canada;
      richardrjohnson81738@gmail.com
\*    Correspondence: gelbart@advancesystems.ca or williamgelbart@gmail.com

**Abstract:** A significant number of medical radioisotopes use solid, often metallic, parent materials. These materials are deposited on a substrate to facilitate the cooling and handling of the target during placing, irradiation, and processing. The processing requires the transfer of the target to a processing area outside the irradiation area. In this new approach the target is processed at the irradiation site for liquid only transport of the irradiated target material to the processing area. The design features common to higher energy production target systems are included in the target station. The target is inclined at 14 degrees to the beam direction. The system has been designed to accept an incident beam of 15 to 16 mm diameter and a beam power between 2 and 5 kW. Thermal modeling is presented for targets of metals and compounds. A cassette of five or 10 prepared targets is housed at the target station as well as a target dissolution assembly. Only the dissolved target material is transported to the chemistry laboratory so that the design does not require additional irradiation area penetrations. This work presents the design, construction, and modeling details of the assembly. A full performance characterization will be reported after the unit is moved to a cyclotron facility for beam related measurements.

**Keywords:** radioisotopes; medical radioisotopes; radioisotope targetry; solid radioisotope targets; radioisotope target processing; medical cyclotrons

---

## 1. Introduction

In solid target irradiation systems the solid target must be transferred to the irradiation position for irradiation and then removed for processing.

There are commercial solid target systems available. For example IBA offers a system called NIRTA [1]. Comecer offers the ALCEO system [2] and ARTMS offers a solid target system [3]. Users have adapted these systems to their needs. For example Carzaniga et al. at the University of Bern reported their developments based on a NIRTA system at the conference in reference [4]. The general features of these systems are that they use disk target bodies with normal beam entrance and transport the irradiated metallic target to the radiochemistry area for subsequent processing.

In the simplest form each target is transferred manually between the hot-cell and the accelerator. More sophisticated systems use some form of mechanical or pneumatic transfer. They are not only more complex and expensive, but require large passages to the accelerator vault or target cave, passages that are difficult to add to existing installations and that can create radiation "leaks".

The system reported here uses a target inclined at 14 degrees to the beam and the target is processed at the irradiation site for liquid transport of the irradiated target material to the radiochemistry area. The new design that is reported here is an evolution of high current target designs adapted to lower current accelerator facilities that may not have the required target or cyclotron vault penetrations.

This self-contained unit automatically places the targets in the irradiation position and at the end of irradiation transfers them to an integral dissolution cell located at the target station itself. Fresh targets—up to 10 in the regular configuration—are contained in a detachable cassette that can be easily and quickly replaced once all the targets are irradiated and processed. In most cyclotron installations the cassette change can coincide with the regular maintenance shutdown.

The advantage of this approach is that only the dissolution liquid and the liquid dissolved material are transferred between the target system and the processing hot-cell. The liquids are moved through a small diameter tube that is easily routed through the existing cable ducts. There is no need for large, dedicated passages for the pneumatic transfer pipes (in the case of pneumatically transferred targets) and no need for a large hot-cell to house the target receiver nor any manual manipulation of the target in the hot-cell; the dissolution liquid is sent directly to the chemistry module.

## 2. Materials and Methods

A small aluminum cage contains all the system's components. Three pneumatic actuators remove a fresh target from the cassette, place it in the irradiation chamber, and insert it in the dissolution vessel at the end of the irradiation. The vessel is equipped with heaters and the temperature of the liquid controlled. The system, indicating the main components, is shown in Figure 1.

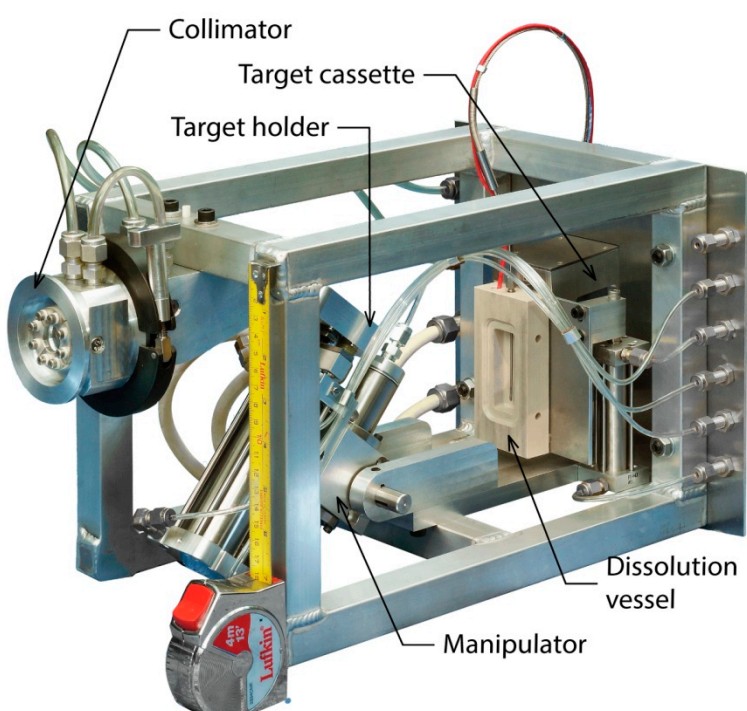

**Figure 1.** Target system assembly.

The solid target substrate is usually made from copper or silver. The face of the target is clad with the irradiated material and cooling channels on the back are designed to dissipate the heat load while maintaining the target face temperature at the desired level. The target is placed at 14 degrees to the beam and the irradiated area forms an ellipse of approximately 13 mm × 52 mm.

The system is designed for a 15 mm–16 mm diameter, roughly circular beam. If no beam focusing elements are present in the beam-line the target location is set to a distance at which the beam diverges to this size. In most small cyclotrons this distance is in the order of less than one meter. To obtain the correct beam coverage on the target material and to get closer to the optimal "top hat" beam cross section, the beam is collimated to 13 mm diameter by a conical, water cooled collimator attached to the front of the system's irradiation chamber; 10% to 15% of the beam power is dissipated in the

collimator. Both the collimator beam current and the target beam current are monitored. To provide accurate target current reading the target and the irradiation chamber form an electrically insulated Faraday cage.

An optional four sector insulated mask (shown in Figure 1) can be installed on the front surface of the collimator. This allows accurate beam positioning, essential in beam-lines with steering/focusing elements and helpful in all instances when beam drifts shift the impact spot. With a well centered on target beam the mask segments will not read any current, and current reading on the sectors will indicate a drift in that direction. The sectors are made out of pure silver and the sector currents are fed-through to the outside via an all-metal seal. Though silver does activate with a 6 h half-life positron from Cd107, and Cd109 auger low energy X-rays with 463 days half-life, the former decays quickly and the latter are very low energy X-rays. Silver has a high heat conductivity and mechanical strength and is a desirable material for collimators that are in the beam periphery.

The angle of the target spreads the beam over a larger area and allows thinner target material cladding (about 25% of the thickness compared to a 90 degrees target). This not only helps with the heat transfer, but allows cladding with materials that cannot easily be deposited in thicker layers. Figure 2 shows the target and the irradiated area dimensions.



**Figure 2.** Solid target, front and back. The **left** figure shows the front irradiated area in grey, the O ring in yellow and the target substrate in orange. The **right** figure shows the cooling channels milled into the substrate and the O ring in yellow. The dimensions are in mm.

The sequence of operation of the system is shown in Figures 3–7.

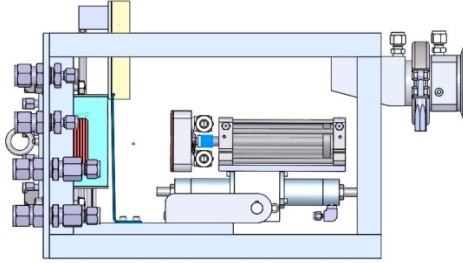

**Figure 3.** Ready for target placing.

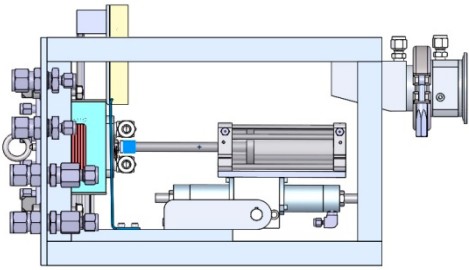

**Figure 4.** Grabbing the target from the cassette.

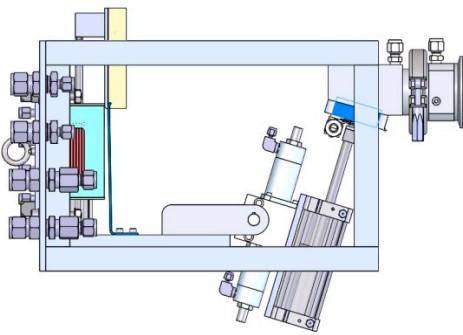

**Figure 5.** The target in irradiation position.

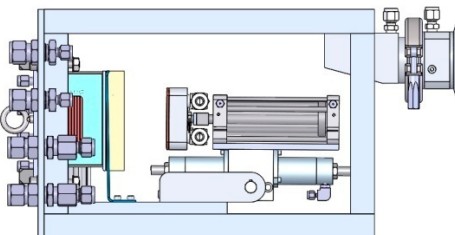

**Figure 6.** The irradiated target ready for dissolution.

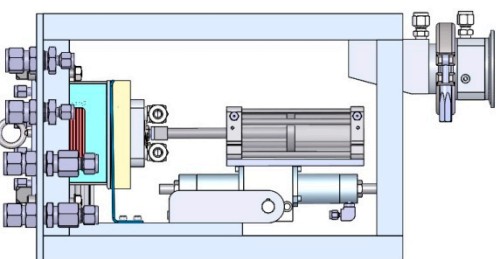

**Figure 7.** The target placed for dissolution.

Fresh targets (up to 10 usually, with a practical limit of 20) are pre-loaded in the detachable target cassette. The cassette snaps in place and can be easily changed in under a minute using a long-reach tool.

Used irradiated targets are released at the end of dissolution and dropped into a shielded container placed under the system and can be left there until the radioactivity levels drop.

In most cases the targets are not reused and the used targets can be stored on the container until disposal time. Figure 8 shows the empty cassette.

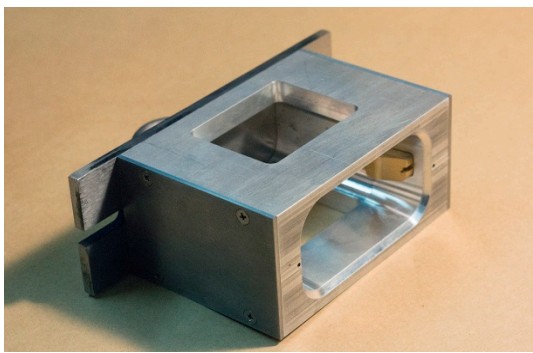

**Figure 8.** The target cassette.

## 3. Results

This solid target irradiation station design is the result of close to forty years of experience building and operating solid targets for radioisotope production and is yet another refinement of a similar system designed and built previously [5–11]. Construction methods and materials were carefully chosen to ensure low activation and to minimize radiation damage to the components. Whenever possible, aluminum, ceramics, and polyimides were used. This design criterion was followed for parts that were manufactured and commercial components such as connectors and fittings only available in other materials were kept in those materials. The only elastomer seals are on the target. These are, of course, used only for one irradiation run. The seals in all the air cylinders are100% graphite thus eliminating the dominant weakness of air actuators in radiation areas since the commercial elastomer seals are exposed to high radiation fields. Water and air tubing are polyurethane. They can serve for a number of years under normal operating conditions.

The target substrate can be supplied in different configurations of the number and size of cooling channels to optimize the cost versus the power requirement. A simplified, cost effective design (Figure 2) is capable of dissipating 2 KW of beam power on target (representing a typical 15 MeV cyclotron delivering 150 μA total beam) with only moderate surface temperature increase. As an example, Figure 9 shows the result of a simulation of the copper target with 60 micron molybdenum cladding and 6 L/min cooling water flow. 150 μA, a 2.3 KW total beam collimated to 2 KW beam on target.

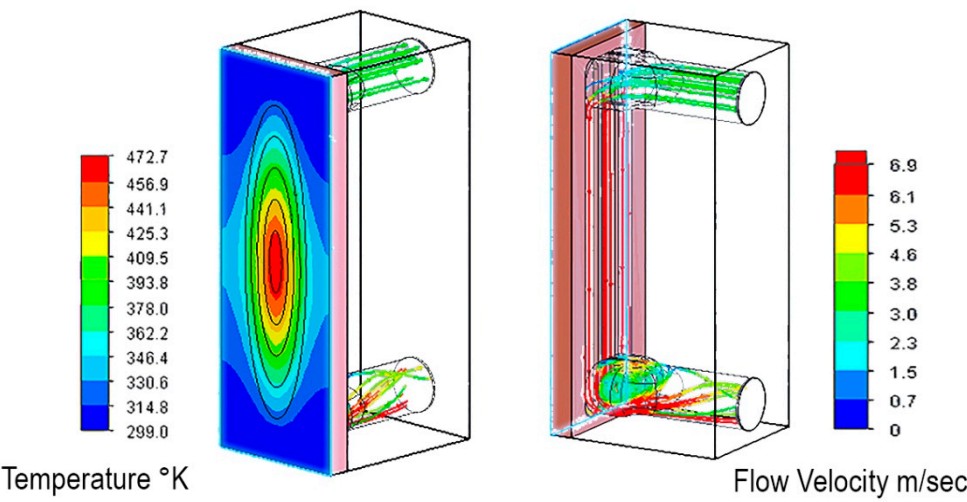

**Figure 9.** Flow and thermal simulation showing the water trajectories and velocity in m/sec (on the **right**) and surface temperature contours in degrees K (on the **left**).

Different cooling channel configurations and higher water flow can increase the power capability up to 5 KW (nine 0.8 mm wide × 1.7 mm deep water channels) with a cost increase of the target fabrication of about 50%. All the water channel designs use turbulent water flow with no cavitation along the channels.

As is the case for all solid targets the "power dissipation capability" depends not only on the supplied water flow and beam distribution on the target face, but on the maximum allowable target surface temperature. This temperature is a function of the target material cladding and the ability of the cladding to withstand this temperature. An excessive temperature manifests itself as mechanical stress flaking, and actual cladding melting. (Note that sputtering of the target material is not possible since there are no energetically ions or electrons to induce the process.) Reducing the target temperature is the important reason for the target inclination to the beam since the beam power is then distributed over a larger area. The thinner cladding also reduces mechanical stress at the boundary of the cladding and substrate. Careful uniform preparation of the cladding reduces mechanical stress as well.

The remaining variable is the power delivered to the cladding so different materials will have different maximum beam currents. In the above examples and ratings a full contact between the cladding and the substrate is assumed. The surface temperature must, of course, be below the melting point of the target material and for low melting point materials the power should be adjusted accordingly by the accelerator operator limiting the beam current.

Special face-grooved solid targets that can support non-metallic target materials were built and tested (Figure 10a). This type of target can be used with materials that can be melted into the grooves (melting point lower than the substrate).

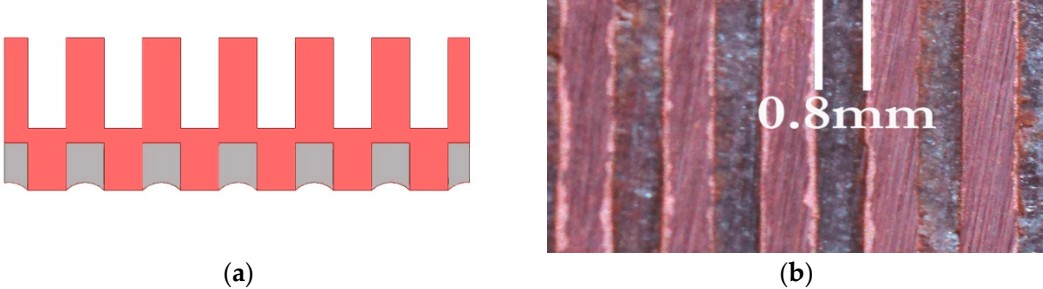

(**a**)                                                                                                          (**b**)

**Figure 10.** (**a**) Grooved target cross-section; the target material is grey while red is the copper substrate. (**b**) Face close-up with fused target material (grey color).

Figure 10b is a close-up of a typical grooved target face with rubidium chloride (MP 718 °C) fused between the grooves for higher energy irradiations and Figure 11 is a representative example of thermal modeling of the central segment of the target (highest heat flux region) for a copper substrate with 0.8 mm wide × 1 mm deep face grooves and nine 0.8 mm wide × 1.7 mm deep water channels. The beam is a 4 KW beam on target and the cooling water flow 10 L/min.

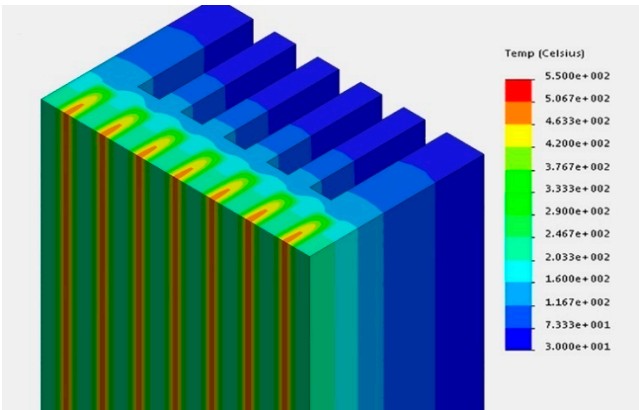

**Figure 11.** Grooved target thermal modeling.

The dissolution cell can be configured for a different target material. It has a heater provision for up to 100 °C heating, and agitation, if needed, is accomplished by inert gas bubbling. A stream selector manifold for solvents and wash liquids is located in the radiochemistry laboratory and delivered to the cell through capillary tubing.

The whole system is small with a footprint of only 350 mm × 200 mm and 240 mm vertical clearance. The system requires a supply of cooling water and compressed air. All target manipulation and processing is done remotely and automatically with interlocks ensuring proper sequences. Control of the process is done by a small, dedicated Programmable Logic Controller (PLC) and can be integrated with the accelerator controls.

## 4. Discussion and Summary

The design of a solid target station that uses targets tilted at 14 degrees to distribute the heat over a larger area, together with at target station target material dissolution after irradiation with subsequent liquid transfer to the radiochemistry laboratory was presented. Heat distribution modeling was conducted for several target substrate geometries. Power limits varied between 2 and 5 kW. The different cooling channel configurations add more cost to the target substrates of up to an additional 50%. The target station has been fabricated and a series of bench tests without radioactivity have begun. The full description of the system and performance measurements with the beam will be reported after the assembly has been moved to a cyclotron facility.

The bench tests will establish the operation limits and confirm the automated procedures. In addition, the bench tests will extend the operational parameters to their limits to study failure mechanisms. Since this system is in a high radiation area and repairs are difficult, a very high meantime between failure is required. For example, the dissolution cell will be tested with non-radioactive materials to confirm the chemical processes for the target. The first two targets to be tested are zinc and molybdenum. Note that the dissolution cell is vertical so that after processing a target the cell must be thoroughly cleaned and dried before the spent target is removed so that no liquids or radioactive materials are released into the area. This is performed automatically as a part of the process cycle.

The target station will finally be moved to a cyclotron facility for hot testing. That location has not been finalized.

## 5. Patents

A United States patent, covering some aspects of the system has been submitted.

**Author Contributions:** Conceptualization, Validation, Analysis, Resources, Writing, Review and editing, and Funding were the responsibility of W.Z.G. and R.R.J.

**Funding:** This research received no external funding.

**Conflicts of Interest:** The authors declare no conflict of interest.

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
