# Peer review of "Solid Target System with In-Situ Target Dissolution"

_instruments, doi:10.3390/instruments3010014_

Round 1

Reviewer 1 Report

1)A short paper describing a versatile and highly needed add-on to existing low-to-medium energy cyclotron targetry. The paper is very short, and only describe the design features, the mechanical functions and some theoretical considerations on its performance under beam. No new or supporting experimental data is given. Results obtained with this target system or similar designs are hidden in some of the references ,9-12, but these are all already published.  No experimental verification is given concerning the feasability and time consumption/activity loss for the "in situ" dissolution process, but this is an important parameter for such a target. Please provide some real data.

2) More details should be given concerning the dissolution cell behaviour: i.e. : what material(s) have been tested, what solution agents ( acids, bases... ) have been tested, what temperatures have been tested,  how is agitation achieved, can gas evolution be handled, what are typical (liquid ) losses (volume), can target and dissolution be efficiently dried before opening, how are the inevitable activity spills when opening the vertical dissolution handled ? 

3) In some places, the manuscript tends to become commercial advertisment material. ( example page 4, line 100). The manuscript should be cleaned for such references. It should be enough to state the number of targets thta have actully been handled in a single casette. 

4) Two set of results on thermal modelling are presented ( figure 9 and figure 11 ), but materials and methods (software, data libraries ) for such calculation not stated. Must be included.

5) The important section on page 5, Line 122-129 on maximum power dissipation capability only mentions temperature as the limiting factor. For sure, other factors will limit this target design. (mechanical stress, cavitation on the water side, sputtering of the target material). Please mention / discuss this, to make the paper m,ore useful to general audience.

6) What is the use of the shown RbCl target ? (fig-10-11) . It can not be the production of Sr-82? 

7) The manuscripts claims that "low activation" materials have been selcted, but at  the same time silver (!) has been stated as the preferred sect  collimator material ( p.2, L.71ff). Please remove or discuss this discrepancy. 

8) Reference is given to a reference 14) ( on page 4, line 107 ) But reference 14 does not exist. Please rectify. 

9) minor spell and grammar :

L. 14 pentrations--> penetrations

L. 22 Bern--> Bern  ( it a city, not a system)

L. 24 is --> are

Author Response

Reviewer 1 comments and response  for small target station manuscript

The authors thank the reviewers for their comments.  In the spirit of the Workshop on Targets and Target Chemistry  we have reported our new design and its fabrication.  The assembly is now undergoing bench testing and it would be appropriate to report those cold tests when they are completed.  Only after the cold tests will the assembly be moved to a cyclotron laboratory. 

1)A short paper describing a versatile and highly needed add-on to existing low-to-medium energy cyclotron targetry. The paper is very short, and only describe the design features, the mechanical functions and some theoretical considerations on its performance under beam. No new or supporting experimental data is given. Results obtained with this target system or similar designs are hidden in some of the references ,9-12, but these are all already published.  No experimental verification is given concerning the feasability and time consumption/activity loss for the "in situ" dissolution process, but this is an important parameter for such a target. Please provide some real data.

The submission is a design report and working results shall be presented at a later workshop.  See introductory comments.

2) More details should be given concerning the dissolution cell behaviour: i.e. : what material(s) have been tested, what solution agents ( acids, bases... ) have been tested, what temperatures have been tested,  how is agitation achieved, can gas evolution be handled, what are typical (liquid ) losses (volume), can target and dissolution be efficiently dried before opening, how are the inevitable activity spills when opening the vertical dissolution handled ? 

The reviewer noted a most important feature of the system.  Can the dissolution cell operate adequately in the target cave environment?  The reagents are stored  outside the irradiation area so that even transfer times of the solvents are important.  Dissolution cell cleaning and drying are important to constrain release of contaminants into the target cave.  The cell performance shall be reported at a later workshop. In many ways this process is similar to the operation of liquid targets.

3) In some places, the manuscript tends to become commercial advertisement material. ( example page 4, line 100). The manuscript should be cleaned for such references. It should be enough to state the number of targets that have actually been handled in a single cassette.  

We have edited out the commercial comments.

4) Two set of results on thermal modelling are presented ( figure 9 and figure 11 ), but materials and methods (software, data libraries ) for such calculation not stated. Must be included.

5) The important section on page 5, Line 122-129 on maximum power dissipation capability only mentions temperature as the limiting factor. For sure, other factors will limit this target design. (mechanical stress, cavitation on the water side, sputtering of the target material). Please mention / discuss this, to make the paper m,ore useful to general audience.

This now is included in lines 144 to 151

6) What is the use of the shown RbCl target ? (fig-10-11) . It can not be the production of Sr-82? 

This example was taken from work for a higher energy (70 MeV) application.  This has been noted in line 163

7) The manuscripts claims that "low activation" materials have been selcted, but at  the same time silver (!) has been stated as the preferred sect  collimator material ( p.2, L.71ff). Please remove or discuss this discrepancy

Yes that is inconsistent.  See mechanical and heat conductivity. Comments in lines 81 to 84 

8) Reference is given to a reference 14) ( on page 4, line 107 ) But reference 14 does not exist. Please rectify. 

Ok  removed ref 14

9) minor spell and grammar :

L. 14 pentrations--> penetrations

ok

L. 22 Bern--> Bern  ( it a city, not a system)

ok

L. 24 is --> are

ok

Reviewer 2 Report

General comments

The authors have presented a prototype of a solid target irradiation system that supports in-situ target dissolution to alleviate the need for transporting the irradiated metallic target to the radiochemistry area for post-processing. This is achieved by configuring the position of the target such that it is inclined at 14 degrees relative to the beam and by processing the irradiated target locally at the irradiation site, which enables the subsequent transport of the irradiated liquid-only target material to the radiochemistry area. This feature can be quite useful as it allows the adaptation of high current target designs to lower current accelerator facilities that may not have the required target or cyclotron vault penetrations.

The authors have conducted a thorough presentation of the features of their design and have provided detailed figures explaining the basic components of the unique characteristics of their proposed solid target irradiation system. Nevertheless, the presented figures need to be improved to better illustrate their reported results. Moreover, further quantitative results should be presented demonstrating the actual operation of the system and the quality of the produced targets. Comparisons with quantitative performance of other existing and established solid target systems should be added. The comparison should be made according to international performance metric standards, if they exist. In addition, the current “Discussion” section should be merged with the Results section and a new “Discussion” section should be added where the authors will summarize their major findings, adequately and fairly discuss limitations in their proposed design, how they plan to address them in future and state their conclusions. Furthermore, the syntax in certain statements throughout the abstract and the main manuscript text should be improved to enhance clarity and readability. Further details and editing/revision suggestions can be found in the commented section below:

Specific comments

Abstract: Numerical/quantitative results should be included in the abstract.

Abstract, Line 12, 13, “It is an evolution…. ” The authors should avoid using "it" and instead specify the subject of their concluding statement in the abstract.

Abstract, Line 16, “pentrations”: "penetrations"

Line 35, “the target”: Please remove the article "the".

Line 44, “for a large”: Please remove the article "a"

Line 78, “with of the target material”: The authors should correct the phrase "with of the". I is syntactically wrong and does not make sense.

Figure 2: For clarity, the authors should specify which of the sub-figures (left or right) illustrate the front and back side and indicate which dimensions designate irradiated area and which the target area.

Lines 108-109, “Whenever possible”: To enhance readability, a "comma" mark should be added after "Whenever possible". In addition, the authors should specify when is possible to use these materials and why is not always possible?

Lines 110-111, “are graphite”: The authors should consider revising as follows: "consist of 100% graphite" or “consist mainly of graphite”.

Line 111, “the biggest weakness”: The authors should explicitly specify the "biggest weakness" and briefly justify why it is characterized as such.

Line 112, “are polyurethane”: The authors should avoid the syntax "something is [material/element]" and instead use a term similar to the following "something consists of % of [material/element]". In this case, the following revision is suggested: "tubing consists of 100% polyurethane", or "consists mainly of polyurethane". The same revision should be issued accordingly throughout the manuscript.

Line 113, “in different configuration… channels”: The following revision is suggested: "in different configurations of the cooling channels".

Lien 118, “6 l/min cooling water flow”: The range of cooling water flows in the diagram of Fig. 9 is 0-7 ml/sec, which is different from the reported 6 l/min= 100ml/sec in the text. The authors should revise accordingly or provide an adequate explanation in the manuscript text and figure 9 caption.

Lines 121-122, “with a slight cost increase… fabrication”: The authors should briefly discuss in the (currently missing) discussion section the different possible cooling channels configurations for their proposed configuration and what is the estimated cost increase in terms of percentage for their configuration.

Lines 125-126, “and that ability…. This temperature”: The syntax of this clause should be revised. The current syntax does not make sense. The authors should consider rephrasing this statement.

Line 128, “the power is adjusted”: The following revision should be issued: "the power should be adjusted accordingly". The authors should specify how the power can be adjusted and if this procedure should be done manually or whether the proposed mechanism supports an automated power adjustment.

Discussion section: The current “Discussion” section in the manuscript is actually not a discussion or conclusion section. It should instead be incorporated to the Results section. The authors should also provide then a separate discussion section where they will briefly summarize their findings, discuss limitations and future steps to address them and finally state their concluding remarks.

Figure 10: The red makings above the two red vertical lines in Fig 10 are not visible. The authors should enlarge their size. In addition, the locations of the target face and the target material should be designated in the figure.

Line 146, “operation”: The following revision should be issued: "operations"

Lines 147-148, “proper operation”: The authors should avoid repeating the same term multiple times within a sentence, as is the case with the term "operation" in this statement.

Line 151, “United States… applied for.”: United States patent, covering some aspects of the system, applied for.

Author Response

Reviewer 2 comments  and response  Solid target station

Comments and Suggestions for Authors

General comments

The authors have presented a prototype of a solid target irradiation system that supports in-situ target dissolution to alleviate the need for transporting the irradiated metallic target to the radiochemistry area for post-processing. This is achieved by configuring the position of the target such that it is inclined at 14 degrees relative to the beam and by processing the irradiated target locally at the irradiation site, which enables the subsequent transport of the irradiated liquid-only target material to the radiochemistry area. This feature can be quite useful as it allows the adaptation of high current target designs to lower current accelerator facilities that may not have the required target or cyclotron vault penetrations.

The authors have conducted a thorough presentation of the features of their design and have provided detailed figures explaining the basic components of the unique characteristics of their proposed solid target irradiation system. Nevertheless, the presented figures need to be improved to better illustrate their reported results. Moreover, further quantitative results should be presented demonstrating the actual operation of the system and the quality of the produced targets. Comparisons with quantitative performance of other existing and established solid target systems should be added. The comparison should be made according to international performance metric standards, if they exist. In addition, the current “Discussion” section should be merged with the Results section and a new “Discussion” section should be added where the authors will summarize their major findings, adequately and fairly discuss limitations in their proposed design, how they plan to address them in future and state their conclusions. Furthermore, the syntax in certain statements throughout the abstract and the main manuscript text should be improved to enhance clarity and readability. Further details and editing/revision suggestions can be found in the commented section below:

The authors thank the reviewers for their comments.  In the spirit of the Workshop on Targets and Target Chemistry  we have reported our new design and its fabrication.  The assembly is now undergoing bench testing and it would be appropriate to report those cold tests when they are completed.  Only after the cold tests will the assembly be moved to a cyclotron laboratory.  We have attempted to include more description in the manuscript and anticipate a full performance report at a later workshop.

Abstract: Numerical/quantitative results should be included in the abstract. 

Abstract modified to include more quantitation

Abstract, Line 12, 13, “It is an evolution…. ” The authors should avoid using "it" and instead specify the subject of their concluding statement in the abstract.

Abstract modified

Abstract, Line 16, “pentrations”: "penetrations"

modified

Line 35, “the target”: Please remove the article "the".

removed

Line 44, “for a large”: Please remove the article "a"

modified

Line 78, “with of the target material”: The authors should correct the phrase "with of the". I is syntactically wrong and does not make sense.

modified

Figure 2: For clarity, the authors should specify which of the sub-figures (left or right) illustrate the front and back side and indicate which dimensions designate irradiated area and which the target area.

Description included in the figure caption

Lines 108-109, “Whenever possible”: To enhance readability, a "comma" mark should be added after "Whenever possible". In addition, the authors should specify when is possible to use these materials and why is not always possible?

Another sentence included to comment on the limitations

Lines 110-111, “are graphite”: The authors should consider revising as follows: "consist of 100% graphite" or “consist mainly of graphite”.

Noted and sentences modified

Line 111, “the biggest weakness”: The authors should explicitly specify the "biggest weakness" and briefly justify why it is characterized as such.

Explanation included in sentence

Line 112, “are polyurethane”: The authors should avoid the syntax "something is [material/element]" and instead use a term similar to the following "something consists of % of [material/element]". In this case, the following revision is suggested: "tubing consists of 100% polyurethane", or "consists mainly of polyurethane". The same revision should be issued accordingly throughout the manuscript.

Qualifying “mainly of” included in the sentence

Line 113, “in different configuration… channels”: The following revision is suggested: "in different configurations of the cooling channels".

modified

Lien 118, “6 l/min cooling water flow”: The range of cooling water flows in the diagram of Fig. 9 is 0-7 ml/sec, which is different from the reported 6 l/min= 100ml/sec in the text. The authors should revise accordingly or provide an adequate explanation in the manuscript text and figure 9 caption.

 The Value in the Figure 6 is the flow velocity in m/sec. This being a conjugate heat/flow analysis the water velocity in the cooling channels is solved together with the heat load and the

cooling cooficents.

Lines 121-122, “with a slight cost increase… fabrication”: The authors should briefly discuss in the (currently missing) discussion section the different possible cooling channels configurations for their proposed configuration and what is the estimated cost increase in terms of percentage for their configuration.

This was done, and the increase changed to approximately 50% (this is a general figure and depends on the number of targets fabricated in each batch)

Lines 125-126, “and that ability…. This temperature”: The syntax of this clause should be revised. The current syntax does not make sense. The authors should consider rephrasing this statement.

Syntax changed

Line 128, “the power is adjusted”: The following revision should be issued: "the power should be adjusted accordingly". The authors should specify how the power can be adjusted and if this procedure should be done manually or whether the proposed mechanism supports an automated power adjustment.

Sentence modified to indicate cyclotron operator does the change

Discussion section: The current “Discussion” section in the manuscript is actually not a discussion or conclusion section. It should instead be incorporated to the Results section. The authors should also provide then a separate discussion section where they will briefly summarize their findings, discuss limitations and future steps to address them and finally state their concluding remarks.

Noted and ending modified

Figure 10: The red makings above the two red vertical lines in Fig 10 are not visible. The authors should enlarge their size. In addition, the locations of the target face and the target material should be designated in the figure.

Enlarged and material indicated

Line 146, “operation”: The following revision should be issued: "operations"

Change made

Lines 147-148, “proper operation”: The authors should avoid repeating the same term multiple times within a sentence, as is the case with the term "operation" in this statement.

 Change made

Line 151, “United States… applied for.”: United States patent, covering some aspects of the system, applied for.

Change made

Round 2

Reviewer 1 Report

"In the spirit of the WTTC", I can recommend this manuscript in the current redacted form as part of a special issue conntaining the proceedings from the WTTC17 in Coimbra. Only in this context, the preliminary nature of the manuscript and the abscence of experimental results can be justified.

The short, factual changes I have asked for have been made to my satisfaction.

Author Response

Comments and response for reviewer 1

Comments and Suggestions for Authors

"In the spirit of the WTTC", I can recommend this manuscript in the current redacted form as part of a special issue conntaining the proceedings from the WTTC17 in Coimbra. Only in this context, the preliminary nature of the manuscript and the abscence of experimental results can be justified.

The short, factual changes I have asked for have been made to my satisfaction.

The authors thank reviewer 1 for the comments.

Reviewer 2 Report

General Comments

The authors have addressed most of the major comments raised in the previous review cycle. However, the current version still lacks the reporting of important quantitative performance results. It is understood that performance would be primarily evaluated after installation of the system. The authors should provide all possible performance measurements currently available both in the abstract and the main manuscript and explicitly state in the discussion section that further performance metrics measurements cannot be reported until the target system is installed on a site. Furthermore, a few additional minor issues needs to be resolved before further consideration can be made, as explained in the specific comment section below.

Specific comments

Abstract: In the abstract, the authors should include all major numerical/quantitative results currently available and mentioned in the main manuscript text.

Line 12, “It is an evolution of high current…”: The authors should avoid using "it" and instead specify the subject of their concluding statememt in the abstract.

Line 14, “irradiation area pentrations”: The typo error should be corrected as follows: "penetrations".

Line 35, “the required the target”: Please remove the article "the" before the term “target”.

Line 44, “for a large, dedicate passages”: Please remove the article "a" before the term “large”.

Line 78, “with of the target material”: Please correct the phrase "with of the".

Figure 2: For clarity, the authors should specify which of the sub-figures (left or right) illustrate the front and back side and indicate which dimensions designate irradiated area and which the target area.

Line 109, “possible aluminium”: To enhance readability, a "comma" mark should be added after "Whenever possible". In addition, the authors should  specify when is possible to use these materials and why is not always possible?

Lines 110-111, “are graphite”: The authors should consider revising as follows: "consist of 100% graphite".

Line 111, “the biggest weakness of”: The authors should explicitly specify the "biggest weakness" and briefly justify why it is characterized as such.

Line 112, “are polyurethane”: The authors should avoid the syntax "something is [material/element]" and instead use a  term similar to the following "something consists of % of [material/element]". In this case, the following revision is suggested: "tubing consists of 100% polyurethane", or "consists mainly of polyurethane". The same revision should be issued accordingly throughout the manuscript.

Line 113, “in different configuration of the cooling channels”: The following revision is suggested: "in different configurations of the cooling channels".

Line 118, “and 6 l/min cooling water flow”: The range of flows in the diagram is 0-7 ml/sec, which is different from the 6 l/min= 100ml/sec. The authors should revise accordingly or provide an adequate explanation in the manuscript.

Figure 9: The units in the color bars on the left of Fig 9 are not visible clearly. They should be enlarged. Which metric (temperature or flow velocity) is depicted on the left map and which on the right map? This piece of info should be clearly provided in the figure caption.

Lines 121-122, up to 5kW with a slight cost increase of the target fabrication”: The authors should briefly discuss in the (currently missing) discussion section the different possible cooling channels configurations for their proposed configuration and what is the estimated cost increase in terms of percentage for their configuration.

Lines 125-126, “and that ability… this temperature.”: The syntax of this clause should be revised. The current syntax does not make sense. The authors should consider rephrasing this statement.

Line 188, “the power is adjusted”: The following revision should be issued: "the power should be adjusted accordingly". The authors should specify how the power can be adjusted and if this procedure should be done manually or whether the proposed mechanism supports an automated power adjustment.

Discussion section: This is not a discussion or conclusion sections. It should be incorporated to the Results section. The authors should also provide then a separate discussion section where they will briefly summarize their findings, discuss limitations and future steps to address them and finally state their concluding remarks.

Figure 10: The red makings above the two red vertical lines in Fig 10 are not visible. The authors should enlarge their size. In addition, the locations of the target face and the target material should be designated in the figure.

Line 146, “All operation of…”: The authors should correct the typo error as follows: "All operations of…".

Lines 147-148, “ensuring proper operation.”: The authors should avoid repeating the same term multiple times within a sentence, as is the case with the term "operation" in this statement.

Line 151, “United States patent, … , applied for.”: The syntax of this clause should be corrected to become plausible and easy to comprehend.

Author Response

Reviewer 2 comments and response    1 Feb 2019

Comments and Suggestions for Authors

General Comments

The authors have addressed most of the major comments raised in the previous review cycle. However, the current version still lacks the reporting of important quantitative performance results. It is understood that performance would be primarily evaluated after installation of the system. The authors should provide all possible performance measurements currently available both in the abstract and the main manuscript and explicitly state in the discussion section that further performance metrics measurements cannot be reported until the target system is installed on a site. Furthermore, a few additional minor issues needs to be resolved before further consideration can be made, as explained in the specific comment section below.

We have included more quantitative detail in the abstract and explicitly stated that performance will be reported when beam tests are made.

 Note that the referee line numbers were somewhat inconsistent with the edited manuscript version 2 and of  course now the line numbers are again  different in this new revision.

Specific comments

Abstract: In the abstract, the authors should include all major numerical/quantitative results currently available and mentioned in the main manuscript text.

Line 12, “It is an evolution of high current…”: The authors should avoid using "it" and instead specify the subject of their concluding statememt in the abstract.  This is now on line 43

 Line 14, “irradiation area pentrations”: The typo error should be corrected as follows: "penetrations".  This  is on line 45

Line 35, “the required the target”: Please remove the article "the" before the term “target”.

This is on line 44

Line 44, “for a large, dedicate passages”: Please remove the article "a" before the term “large”.

This is on line 54

Line 78, “with of the target material”: Please correct the phrase "with of the".

Figure 2: For clarity, the authors should specify which of the sub-figures (left or right) illustrate the front and back side and indicate which dimensions designate irradiated area and which the target area. Figure caption now addresses this

Line 109, “possible aluminium”: To enhance readability, a "comma" mark should be added after "Whenever possible". In addition, the authors should  specify when is possible to use these materials and why is not always possible?

Now line 120   

Lines 110-111, “are graphite”: The authors should consider revising as follows: "consist of 100% graphite". explanation  included now on line 123

Line 111, “the biggest weakness of”: The authors should explicitly specify the "biggest weakness" and briefly justify why it is characterized as such.

See line 125

Line 112, “are polyurethane”: The authors should avoid the syntax "something is [material/element]" and instead use a  term similar to the following "something consists of % of [material/element]". In this case, the following revision is suggested: "tubing consists of 100% polyurethane", or "consists mainly of polyurethane". The same revision should be issued accordingly throughout the manuscript.

See line 127

Line 113, “in different configuration of the cooling channels”: The following revision is suggested: "in different configurations of the cooling channels".

See line 129

Line 118, “and 6 l/min cooling water flow”: The range of flows in the diagram is 0-7 ml/sec, which is different from the 6 l/min= 100ml/sec. The authors should revise accordingly or provide an adequate explanation in the manuscript.

The diagram indicates the water velocity in m/sec (not flow). More details are now included in text and a note made in figure caption.

Figure 9: The units in the color bars on the left of Fig 9 are not visible clearly. They should be enlarged. Which metric (temperature or flow velocity) is depicted on the left map and which on the right map? This piece of info should be clearly provided in the figure caption.

 Image and text enlarged and more information in the caption added.

Lines 121-122, up to 5kW with a slight cost increase of the target fabrication”: The authors should briefly discuss in the (currently missing) discussion section the different possible cooling channels configurations for their proposed configuration and what is the estimated cost increase in terms of percentage for their configuration.

 See line 141

Lines 125-126, “and that ability… this temperature.”: The syntax of this clause should be revised. The current syntax does not make sense. The authors should consider rephrasing this statement.

See line 145 … 

Line 188, “the power is adjusted”: The following revision should be issued: "the power should be adjusted accordingly". The authors should specify how the power can be adjusted and if this procedure should be done manually or whether the proposed mechanism supports an automated power adjustment.

See line 156

Discussion section: This is not a discussion or conclusion sections. It should be incorporated to the Results section. The authors should also provide then a separate discussion section where they will briefly summarize their findings, discuss limitations and future steps to address them and finally state their concluding remarks.

See line 188 on.

Figure 10: The red makings above the two red vertical lines in Fig 10 are not visible. The authors should enlarge their size. In addition, the locations of the target face and the target material should be designated in the figure.

New figure 10

Line 146, “All operation of…”: The authors should correct the typo error as follows: "All operations of…".

 Line 182

Lines 147-148, “ensuring proper operation.”: The authors should avoid repeating the same term multiple times within a sentence, as is the case with the term "operation" in this statement.

 line 180

Line 151, “United States patent, … , applied for.”: The syntax of this clause should be corrected to become plausible and easy to comprehend.

see line 210

Submission Date

27 December 2018

Date of this review

01 Feb 2019 03:08:4
